# A Numerical Evaluation of SIFs of 2-D Functionally Graded Materials by Enriched Natural Element Method

## Jin-Rae Cho

Department of Naval Architecture and Ocean Engineering, Hongik University, Jochiwon, Sejong 30016, Korea; jrcho@hongik.ac.kr; Tel.: +82-44-860-2546

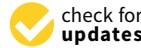

**Featured Application: Prediction of crack propagation of functionally graded materials (FGMs).**

**Abstract:** This paper presents the numerical prediction of stress intensity factors (SIFs) of 2-D inhomogeneous functionally graded materials (FGMs) by an enriched Petrov-Galerkin natural element method (PG-NEM). The overall trial displacement field was approximated in terms of Laplace interpolation functions, and the crack tip one was enhanced by the crack-tip singular displacement field. The overall stress and strain distributions, which were obtained by PG-NEM, were smoothened and improved by the stress recovery. The modified interaction integral $\widetilde{M}^{(1,2)}$ was employed to evaluate the stress intensity factors of FGMs with spatially varying elastic moduli. The proposed method was validated through the representative numerical examples and the effectiveness was justified by comparing the numerical results with the reference solutions.

**Keywords:** enhanced PG-NEM; functionally graded material (FGM); stress intensity factor (SIF); modified interaction integral

## 1. Introduction

In the late 1980s, a new material concept called functionally graded material (FGM) was proposed to resolve the inherent problem of traditional lamination-type composites [1]. The sharp material discontinuity across the layer interface causes the stress concentration, which may trigger the layer delamination. This crucial stress concentration can be significantly minimized by inserting a graded layer between two distinct homogeneous material layers [2,3]. This is because the material discontinuity completely disappears according to the material composition gradation, where the constituent particles of two base materials are mixed up in a random microstructure within a graded layer to maximize the desired performance [4–6]. Naturally, a functionally graded material is an inhomogeneous material, with spatially non-uniform material properties characterized by continuity and functionality. In addition to the suppression of stress concentration, the material concept of FGM rapidly and continuously spread throughout engineering and fields [7–10].

Early research efforts were concentrated on material characterization, fabrication, modeling, and analysis [1,11,12]. This was because the mechanical behaviors of FGMs are governed by the geometric dimensions and orientation, microstructure, and volume fractions of constituent particles. Recently, however, the concern toward the crack problems has increased because the structural failure of FGMs is dominated by micro-cracking [7,13,14]. In this regard, an accurate numerical prediction of stress intensity factors and the crack propagation has been an essential research subject [15,16]. For these subjects, an accurate reproduction of the $\frac{1}{\sqrt{r}}$ singularity in the near-tip stress field in highly heterogeneous media becomes a key task [17–20].

To evaluate the stress intensity factors of FGMs with cracks, one can consider the use of the well-known $J-$ or $M-$integral methods. However, these conventional indirect integral methods cannot reflect the spatially varying material properties of FGMs. The studies on the fracture mechanics of inhomogeneous bodies were initiated in the 1970~80s by assuming the spatially varying elastic modulus as an exponential function [17,21]. The standard integral methods were modified and/or refined to reflect the spatial non-uniformity of material properties by subsequent investigators. The most works were made by utilizing the finite element method [15,22–25]. However, since late 1990s, the employment of meshless methods to crack problems of inhomogeneous bodies has been actively advanced, particularly for the computation of SIFs by the modified $M-$integral method [26,27]. Here, the extension of the natural element method (NEM) is worth noting, even though it was restricted to 2-D homogenous material [28].

The natural element method was introduced to overcome the demerits of conventional meshless methods [29], the difficulty in enforcing the displacement constraint and the numerical integration. The Laplace interpolation functions in NEM strictly obey the Kronecker delta property so that the imposition of displacement constraint becomes easy. In addition, Delaunay triangles, which were produced during the process for introducing Laplace interpolation functions, also serve as a background mesh for the numerical integration. In particular, PG-NEM can further improve the numerical integration accuracy by maintaining the consistency between the Delaunay triangle and the integration region [30]. Although Laplace interpolation functions provide the high smoothness of $C^1$-continuity, there is still room for further improvement in capturing the high stress singularity at the crack tip.

In this context, this paper introduces an enriched PG-NEM to explore whether and how much the enrichment of interpolation function increases the prediction reliability of stress intensity factors for FGMs. The validity of enrichment was reported for homogeneous materials [31,32], but it was rarely reported for inhomogeneous materials. The trial function is enriched by the asymptotic displacement fields of mode I and II, and the approximated overall stress field is enhanced by the patch recovery technique. The proposed method was validated through the illustrative numerical examples and its effectiveness was quantitatively evaluated.

## 2. 2-D Inhomogeneous Cracked Bodies

### 2.1. Linear Elasticity of 2-D Cracked Bodies

Figure 1 represents a 2-D linearly elastic isotropic inhomogeneous material with an edge crack which is contained within a bounded domain $\Omega \in \Re^2$ with the boundary $\partial\Omega = \overline{\Gamma_D \cup \Gamma_N \cup \Gamma_c}$. Here, $\Gamma_D$ and $\Gamma_N$ indicate the displacement and force boundary regions, and $\Gamma_c = \overline{\Gamma_c^+ \cup \Gamma_c^-}$ denotes the traction-free crack surface. As a representative non-homogeneous material, FGMs are characterized by the spatially varying elastic modulus $E$ and Poisson's ratio $v$ over the bounded domain $\Omega$. For the mathematical description purpose, two Cartesian co-ordinate systems are employed, $\{x, y\}$ for the 2-D linear elasticity problem and $\{x', y'\}$ for the SIF evaluation of crack respectively. Assuming the crack surface is traction-free and neglecting the body force $\boldsymbol{b}$, then the displacement field $\boldsymbol{u}(\boldsymbol{x})$ in the Cartesian coordinate system $\{x, y\}$ is subjected to the equilibrium equations

$$\nabla \cdot \boldsymbol{\sigma} = 0 \quad in \quad \Omega \tag{1}$$

with the displacement constraint

$$\boldsymbol{u} = \hat{\boldsymbol{u}} \quad on \quad \Gamma_D \tag{2}$$

and the force boundary condition

$$\boldsymbol{\sigma} \cdot \boldsymbol{n} = \begin{cases} \hat{\boldsymbol{t}} & on \quad \Gamma_N \\ 0 & on \quad \Gamma_c^\pm \end{cases} \tag{3}$$

Here, $\sigma$ are the Cauchy stresses, $n$ the outward unit vector normal to $\partial\Omega$, and $\hat{t}$ the contour traction. When the displacement and strains are assumed to be small, the Cauchy strain $\varepsilon$ is constituted to the Cauchy stress $\sigma$ via the $(3 \times 2)$ gradient-like operator $L$ such that

$$\varepsilon = \varepsilon(u) = Lu \tag{4}$$

Letting $D$ be the constitutive tensor, the stresses and strains are constituted by

$$\sigma = D : \varepsilon \tag{5}$$

Note that the displacement, strains, and stresses are calculated based on the co-ordinate system $\{x, y\}$ and transformed into the co-ordinate system $\{x', y'\}$.

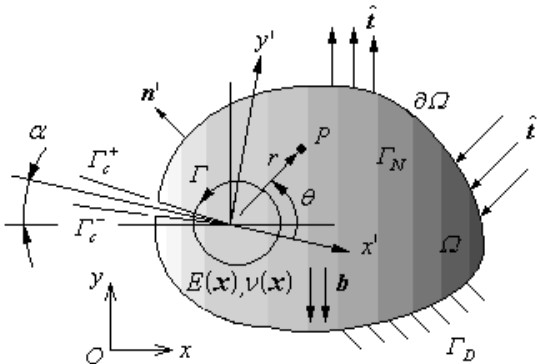

**Figure 1.** An inhomogeneous isotropic body with an edge crack.

For a homogeneous cracked body, the energy release rate per unit crack propagation along the $x'-$axis can be estimated by the path-independent $J-$integral defined by

$$J = \int_{\Gamma}\left(W\delta_{1j} - \sigma_{ij}\frac{\partial u_i}{\partial x'_1}\right)n'_j ds \tag{6}$$

using the indicial notation (i.e., $x'_1 = x'$ and $x'_2 = y'$), the Dirac delta function $\delta_{1j}$, and the strain-energy density $W = \sigma \cdot \varepsilon/2 = \varepsilon_{ij}D_{ijkl}\varepsilon_{kl}/2$. Here, $\Gamma$ indicates an arbitrary closed path, which surrounds the crack tip counterclockwise. As shown in Figure 2, it is expanded to $C = \Gamma + \Gamma_c^- + \Gamma_c^+ + \Gamma_o$ in order to generate a grayed donut-type region A, in which a smooth function $q(x)(0 \leq q \leq 1)$ is introduced to recast the integral into an equivalent domain form [33]. The function $q$, called by weighting function, becomes unity on $\Gamma$, zero on $\Gamma_o$, and arbitrary value between 0 and 1 within the grayed donut region A. Then, the above Equation (6) can be expanded as following

$$J = \int_A\left(\sigma_{ij}\frac{\partial u_i}{\partial x'_1} - W\delta_{1j}\right)\frac{\partial q}{\partial x'_j}dA + \int_A\frac{\partial}{\partial x'_j}\left(\sigma_{ij}\frac{\partial u_i}{\partial x'_1} - W\delta_{1j}\right)qdA \tag{7}$$

according to the divergence theorem, together with $n' = -n'_i$ on $\Gamma$ in C. By further expanding the second term on the right hand side, Equation (7) becomes

$$J = \int_A\left(\sigma_{ij}\frac{\partial u_i}{\partial x'_1} - W\delta_{1j}\right)\frac{\partial q}{\partial x'_j}dA + \int_A\left(\frac{\partial \sigma_{ij}}{\partial x'_j}\frac{\partial u_i}{\partial x'_1} + \sigma_{ij}\frac{\partial^2 u_i}{\partial x'_j\partial x'_1} - \sigma_{ij}\frac{\partial \varepsilon_{ij}}{\partial x'_1} - \frac{1}{2}\varepsilon_{ij}\frac{\partial D_{ijkl}}{\partial x'_1}\varepsilon_{kl}\right)qdA \tag{8}$$

But, the second integral on the right-hand side of Equation (8) becomes zero according to the equilibrium (1), compatibility (4), and the material uniformity. Therefore, the $J-$integral for homogeneous materials becomes the area integral defined by

$$J = \int_A \left( \sigma_{ij} \frac{\partial u_i}{\partial x'_1} - W\delta_{1j} \right) \frac{\partial q}{\partial x'_j} dA \tag{9}$$

But, for non-homogeneous materials, the last material derivation term within the second integrand of Equation (8) does not become zero. Therefore, Equation (8) ends up with a more general $\widetilde{J}-$integral [34], which is given by

$$\widetilde{J} = \int_A \left( \sigma_{ij} \frac{\partial u_i}{\partial x'_1} - W\delta_{1j} \right) \frac{\partial q}{\partial x'_j} dA - \int_A \frac{1}{2} \varepsilon_{ij} \frac{\partial D_{ijkl}}{\partial x'_1} \varepsilon_{kl} q dA \tag{10}$$

The last term in Equation (10) becomes extremely small as a contour $\Gamma_0$ shrinks to the crack tip, but its contribution is not negligible when the domain of integration A is reasonably large.

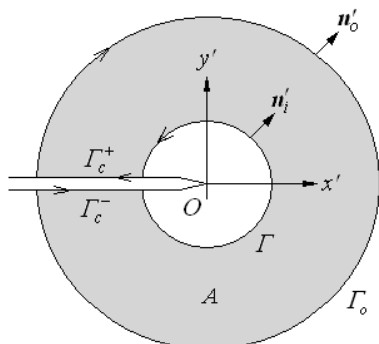

**Figure 2.** An extension of contour $\Gamma$ and a donut-type domain of integration $A$.

## 2.2. Modified Interaction Integral $\widetilde{M}^{(1,2)}$

In order to extract $K_I$ and $K_{II}$ from $J-$integral, one needs to employ the interaction integral, in which two equilibrium states of a cracked body are considered. State 1 is the real equilibrium state of a body that is subjected to the prescribed boundary conditions, while state 2 stands for an auxiliary equilibrium state which would be the asymptotic near-tip fields for modes I or II. Another equilibrium state, called state S, could be established by combining these two states, for which the $\widetilde{J}-$integral in Equation (10) is rewritten as [21]

$$\widetilde{J}^{(S)} = \int_A \left( \left( \sigma_{ij}^{(1)} + \sigma_{ij}^{(2)} \right) \frac{\partial \left( u_i^{(1)} + u_i^{(2)} \right)}{\partial x'_1} - W^{(S)}\delta_{1j} \right) \frac{\partial q}{\partial x'_j} dA + \int_A \frac{\partial}{\partial x'_j} \left( \left( \sigma_{ij}^{(1)} + \sigma_{ij}^{(2)} \right) \frac{\partial \left( u_i^{(1)} + u_i^{(2)} \right)}{\partial x'_1} - W^{(S)}\delta_{1j} \right) q dA \tag{11}$$

with three different strain energy densities, $W^{(1)}$, $W^{(2)}$ and $W^{(1,2)}$, defined by

$$\begin{aligned} W^{(S)} &= \tfrac{1}{2}\left( \sigma_{ij}^{(1)} + \sigma_{ij}^{(2)} \right)\left( \varepsilon_{ij}^{(1)} + \varepsilon_{ij}^{(2)} \right) = \tfrac{1}{2}\sigma_{ij}^{(1)}\varepsilon_{ij}^{(1)} + \tfrac{1}{2}\sigma_{ij}^{(2)}\varepsilon_{ij}^{(2)} + \tfrac{1}{2}\left( \sigma_{ij}^{(1)}\varepsilon_{ij}^{(2)} + \sigma_{ij}^{(2)}\varepsilon_{ij}^{(1)} \right) \\ &= W^{(1)} + W^{(2)} + W^{(1,2)} \end{aligned} \tag{12}$$

By utilizing the equilibrium (1) (i.e., $\partial \sigma_{ij}^{(\cdot)} / \partial x_j = 0$) and the compatibility (4) (i.e., $\varepsilon_{ij}^{(\cdot)} = \partial u_i^{(\cdot)} / \partial x_j$), Equation (11) can be further simplified as

$$
\begin{aligned}
\widetilde{J}^{(S)} &= \int_A \left( \left( \sigma_{ij}^{(1)} + \sigma_{ij}^{(2)} \right) \frac{\partial \left( u_i^{(1)} + u_i^{(2)} \right)}{\partial x'_1} - \left( W^{(1)} + W^{(2)} + W^{(1,2)} \right) \delta_{1j} \right) \frac{\partial q}{\partial x'_j} dA \\
&\quad + \int_A \frac{1}{2} \left( -\varepsilon_{ij}^{(1)} \frac{\partial D_{ijkl}}{\partial x'_1} \varepsilon_{kl}^{(1)} + \sigma_{ij}^{(1)} \frac{\partial \varepsilon_{ij}^{(2)}}{\partial x'_1} - \frac{\partial \sigma_{ij}^{(2)}}{\partial x'_1} \varepsilon_{ij}^{(1)} + \sigma_{ij}^{(2)} \frac{\partial \varepsilon_{ij}^{(1)}}{\partial x'_1} - \frac{\partial \sigma_{ij}^{(1)}}{\partial x'_1} \varepsilon_{ij}^{(2)} \right) q dA \\
&= \widetilde{J}^{(1)} + \widetilde{J}^{(2)} + \widetilde{M}^{(1,2)}
\end{aligned}
\tag{13}
$$

Here,

$$
\widetilde{J}^{(1)} = \int_A \left( \sigma_{ij}^{(1)} \frac{\partial u_i^{(1)}}{\partial x'_1} - W^{(1)} \delta_{1j} \right) \frac{\partial q}{\partial x'_j} dA - \frac{1}{2} \int_A \varepsilon_{ij}^{(1)} \frac{\partial D_{ijkl}}{\partial x'_1} \varepsilon_{kl}^{(1)} q dA
\tag{14}
$$

$$
\widetilde{J}^{(2)} = \int_A \left( \sigma_{ij}^{(2)} \frac{\partial u_i^{(2)}}{\partial x'_1} - W^{(2)} \delta_{1j} \right) \frac{\partial q}{\partial x'_j} dA
\tag{15}
$$

denote the $\widetilde{J}$−integrals for state 1 and state 2 respectively, and

$$
\widetilde{M}^{(1,2)} = \int_A \left( \sigma_{ij}^{(1)} \frac{\partial u_i^{(2)}}{\partial x'_1} + \sigma_{ij}^{(2)} \frac{\partial u_i^{(1)}}{\partial x'_1} - W^{(1,2)} \delta_{1j} \right) \frac{\partial q}{\partial x'_j} dA + \int_A \frac{1}{2} \left( \sigma_{ij}^{(1)} \frac{\partial \varepsilon_{ij}^{(2)}}{\partial x'_1} - \frac{\partial \sigma_{ij}^{(2)}}{\partial x'_1} \varepsilon_{ij}^{(1)} + \sigma_{ij}^{(2)} \frac{\partial \varepsilon_{ij}^{(1)}}{\partial x'_1} - \frac{\partial \sigma_{ij}^{(1)}}{\partial x'_1} \varepsilon_{ij}^{(2)} \right) q dA
\tag{16}
$$

indicates the modified interaction integral. All the quantities in Equation (16) are evaluated based on a coordinate system aligned to the crack tip, and the identification of domain of integration $A$ and the weighting function $q(x)$ will be described in details in Section 3.2.

Since two $\widetilde{J}$−integrals for inhomogeneous cracked bodies which are subject to mixed-mode loading also represent the energy release rates, one can obtain

$$
\widetilde{J}^{(1)} = \frac{1}{\overline{E}_{tip}} \left( K_I^{(1)^2} + K_{II}^{(1)^2} \right)
\tag{17}
$$

$$
\widetilde{J}^{(2)} = \frac{1}{\overline{E}_{tip}} \left( K_I^{(2)^2} + K_{II}^{(2)^2} \right)
\tag{18}
$$

And

$$
\widetilde{J}^{(S)} = \widetilde{J}^{(1)} + \widetilde{J}^{(2)} + \frac{2}{\overline{E}_{tip}} \left( K_I^{(1)} K_I^{(2)} + K_{II}^{(1)} K_{II}^{(2)} \right)
\tag{19}
$$

where $\overline{E}_{tip}$ at the crack tip is $E_{tip}$ for plane stress, while it is $E_{tip} / \left( 1 - v^2 \right)$ for plane strain. By equating Equation (13) with Equation (19), one can obtain

$$
\widetilde{M}^{(1,2)} = \frac{2}{\overline{E}_{tip}} \left( K_I^{(1)} K_I^{(2)} + K_{II}^{(1)} K_{II}^{(2)} \right)
\tag{20}
$$

In 2-D linear fracture mechanics, the closed-form near-tip displacement fields are available for mode I and mode II [35]. The stress intensity factor $K_I^{(1)}$ for mode I can be obtained by letting state 2 be the pure mode-I asymptotic field (i.e., $K_I^{(2)} = 1$ and $K_{II}^{(2)} = 0$):

$$
M^{(1, \text{ModeI})} = \frac{2}{\overline{E}_{tip}} K_I^{(1)}
\tag{21}
$$

In a similar manner, the stress intensity factor $K_{II}$ for mode-II can be also determined.

### 3. Enriched Petrov-Galerkin Natural Element Method

*3.1. Enriched NEM Approximation*

The boundary value problem expressed by Equations (1)–(3) is converted to the weak form according to the virtual work principle: Find $u(x)$ such that

$$\int_\Omega \varepsilon(v) : \sigma(u)\, d\Omega = \int_{\Gamma_N} \hat{t} \cdot v\, ds \tag{22}$$

for all the test displacements $v(x)$ in the Cartesian coordinate system $\{x, y\}$. Next, a non-convex NEM grid $\mathfrak{I}_{NEM}$, shown in Figure 3a, is constructed using $N$ nodes and a number of Delaunay triangles. Then, for a given NEM grid, trial and test displacement fields $u(x)$ and $v(x)$ for PG-NEM approximation are interpolated as

$$u_h(x) = \sum_{J=1}^{N} \bar{u}_J \phi_J(x) + k_1 \begin{bmatrix} Q_{11}(x) \\ Q_{12}(x) \end{bmatrix} + k_2 \begin{bmatrix} Q_{21}(x) \\ Q_{22}(x) \end{bmatrix} \tag{23}$$

$$v_h(x) = \sum_{I=1}^{N} \bar{v}_I \Psi_I(x) + \lambda_1 \begin{bmatrix} Q_{11}(x) \\ Q_{12}(x) \end{bmatrix} + \lambda_2 \begin{bmatrix} Q_{21}(x) \\ Q_{22}(x) \end{bmatrix} \tag{24}$$

with Laplace interpolation functions $\phi_J(x)$ represented in Figure 3b. Here, $\psi_I(x)$ are constant-strain FE basis functions, which are defined over Delaunay triangles.

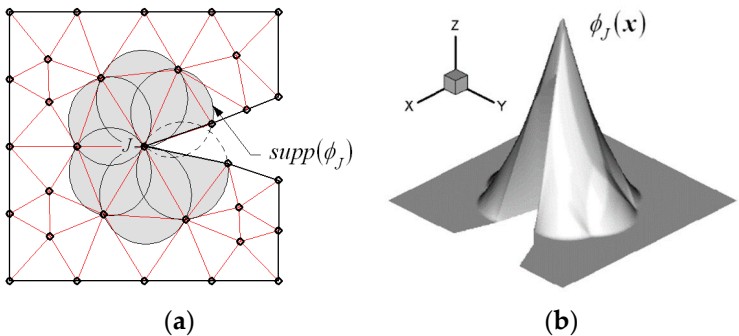

(a)             (b)

**Figure 3.** (a) Non-convex NEM grid $\mathfrak{I}_{NEM}$, (b) Laplace interpolation function $\phi_J(x)$ at the crack tip.

In addition, $k_1$ and $k_2$ are two unknown constants associated with a crack, and $Q_{1\alpha}(x)$ and $Q_{2\alpha}(x)$ represent the near-tip singular displacement fields given by

$$Q_{11}(x) = \frac{1}{2\mu} \sqrt{\frac{r}{2\pi}} \cos\left(\frac{\theta}{2}\right)\left[\kappa - 1 + 2\sin^2\left(\frac{\theta}{2}\right)\right] \tag{25}$$

$$Q_{12}(x) = \frac{1}{2\mu} \sqrt{\frac{r}{2\pi}} \sin\left(\frac{\theta}{2}\right)\left[\kappa + 1 - 2\cos^2\left(\frac{\theta}{2}\right)\right] \tag{26}$$

$$Q_{21}(x) = \frac{1}{2\mu} \sqrt{\frac{r}{2\pi}} \sin\left(\frac{\theta}{2}\right)\left[\kappa + 1 + 2\cos^2\left(\frac{\theta}{2}\right)\right] \tag{27}$$

$$Q_{22}(x) = \frac{-1}{2\mu} \sqrt{\frac{r}{2\pi}} \cos\left(\frac{\theta}{2}\right)\left[\kappa - 1 - 2\sin^2\left(\frac{\theta}{2}\right)\right] \tag{28}$$

Referring to Figure 1, $r$ is the radial distance from the crack tip, while $\theta$ is the angle from the $x'$−axis. Meanwhile, $\mu$ indicates the shear modulus, and $\kappa$ is the Kolosov constant given by

$$\kappa = \begin{cases} 3 - 4\nu & \text{for plane strain} \\ (3 - \nu)/(1 + \nu) & \text{for plane stress} \end{cases} \tag{29}$$

with $\nu$ being the Poisson's ratio. From Equation (23), the overall nodal coefficients $\bar{u}_J$ are defined by

$$\bar{u}_J(x) = u_h(x_J) - k_1 \begin{bmatrix} Q_{11}(x_J) \\ Q_{12}(x_J) \end{bmatrix} - k_2 \begin{bmatrix} Q_{21}(x_J) \\ Q_{22}(x_J) \end{bmatrix} \tag{30}$$

and the overall stress and strain fields corresponding to $\bar{u}_J$ are recovered by the global recovery technique. Meanwhile, the original essential boundary condition (2) is modified as

$$\hat{u} = \hat{u} - k_1 \begin{bmatrix} Q_{11}(\hat{x}) \\ Q_{12}(\hat{x}) \end{bmatrix} - k_2 \begin{bmatrix} Q_{21}(\hat{x}) \\ Q_{22}(\hat{x}) \end{bmatrix}, \quad \hat{x} \quad on \quad \Gamma_D \tag{31}$$

Substituting Equations (23) and (24) into Equation (22), together with Equations (4) and (5), ends up with

$$\begin{bmatrix} K_{IJ} & K_{I1} & K_{I2} \\ K_{I1}^T & K_{11} & K_{12} \\ K_{I2}^T & K_{12}^T & K_{22} \end{bmatrix} \begin{Bmatrix} \bar{u}_J \\ k_1 \\ k_2 \end{Bmatrix} = \begin{Bmatrix} F_I \\ f_1 \\ f_2 \end{Bmatrix} \tag{32}$$

In which the global stiffness matrices $K_{IJ}$ and load vectors $F_I$ are defined by

$$K_{IJ} = \int_{\Omega} B_I^T D B_J d\Omega, \quad F_I = \int_{\Gamma_N} \Phi_I^T \hat{t} \, ds, \qquad I, J = 1, 2, \cdots, N \tag{33}$$

where

$$B_I^T = \begin{bmatrix} \phi_{I,x} & 0 & \phi_{I,y} \\ 0 & \phi_{I,y} & \phi_{I,x} \end{bmatrix} \tag{34}$$

$$D = \frac{E}{(1+\nu)(1-2\nu)} \begin{bmatrix} 1-\nu & \nu & 0 \\ \nu & 1-\nu & 0 \\ 0 & 0 & \frac{1-2\nu}{2} \end{bmatrix} \text{ for plane strain} \tag{35}$$

Meanwhile, the enriched stiffness matrices $K_{\alpha\beta}$, load vectors $f_{\alpha}$, and the interface matrices $K_{I\alpha}$ are defined by, respectively

$$K_{\alpha\beta} = \int_{\Omega} \hat{B}_{\alpha}^T D \hat{B}_{\beta} d\Omega, \quad f_{\alpha} = \int_{\Gamma_N} \hat{\phi}_{\alpha}^T \hat{t} \, ds, \quad \alpha, \beta = 1, 2 \tag{36}$$

$$K_{I\alpha} = \int_{\Omega} B_I^T D \hat{B}_{\alpha} d\Omega \tag{37}$$

with

$$\hat{\phi}_{\alpha}^T = [Q_{\alpha 1}(x), \, Q_{\alpha 2}(x)] \tag{38}$$

$$\hat{B}_{\alpha} = \begin{bmatrix} Q_{\alpha\alpha,x}(x) \\ Q_{\alpha\beta,y}(x) \\ Q_{\alpha\alpha,y}(x) + Q_{\alpha\beta,y}(x) \end{bmatrix}, \quad \alpha\alpha = \text{no sum} \tag{39}$$

### 3.2. Numerical Implementation of $\widetilde{M}^{(1,2)}$

Figure 4 schematically represents the identification of domain of integration $A$ and the definition of weighting function $q(x)$, which are prerequisite for the numerical implementation of $\widetilde{M}^{(1,2)}$ in Eq. (6). For these two things, a circle of the radius $r_{int}$, which is originated at the crack tip, is first imaginarily drawn on the NEM grid. Then, the nodal values of $q(x)$ are assigned based on this circle, such that unity is specified to all the interior nodes while zero to the outer remaining nodes. Then, referring to Figure 2, the boundary of a square composed of eight darkened Delaunay triangles serves as an internal path Γ. On the other hand, the grayed region outside the internal path Γ automatically becomes a domain of integration $A$. According to the properties of Laplace interpolation functions [29], the defined weighting function $q(x)$ becomes unity within a set of darkened Delaunay triangles while it has the values between zero and unity within the grayed domain of integration $A$.

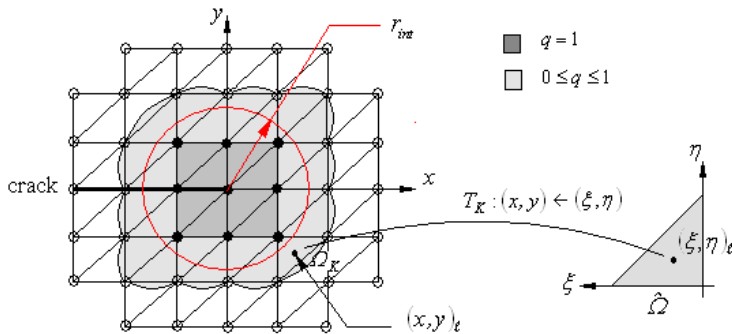

**Figure 4.** A crack tip-originated circle and a graded domain of integration $A$.

Both the weighting function $q(x)$ and the derivative $\partial q/\partial x_j$ become zero outside the domain of integration $A$. Furthermore, the derivative $\partial q/\partial x_j$ vanishes within the darkened squares. So, only the grayed Delaunay triangles are taken for the modified interaction integral $\widetilde{M}^{(1,2)}$ such that

$$\widetilde{M}^{(1,2)} = \sum_{K=1}^{M_G} \widetilde{M}_K^{(1,2)} \tag{40}$$

Here, $M_G$ indicates the total number of grayed Delaunay triangles, while $\widetilde{M}_K^{(1,2)}$ stand for the triangle-wise modified interaction integrals, which are defined by

$$\widetilde{M}_K^{(1,2)} = \sum_{\ell=1}^{INT} \left\{ \left( \left[ \sigma_{ij}^{(1)} \frac{\partial u_i^{(2)}}{\partial x_1} + \sigma_{ij}^{(2)} \frac{\partial u_i^{(1)}}{\partial x_1} - W^{(1,2)} \delta_{1j} \right]_{x_\ell} \frac{\partial q}{\partial x_j} \bigg|_{x_\ell} w_\ell |J|_{x_\ell} \right. \\ \left. + \frac{1}{2} \left[ \sigma_{ij}^{(1)} \frac{\partial \varepsilon_{ij}^{(2)}}{\partial x_1} - \frac{\partial \sigma_{ij}^{(2)}}{\partial x'_1} \varepsilon_{ij}^{(1)} + \sigma_{ij}^{(2)} \frac{\partial \varepsilon_{ij}^{(1)}}{\partial x_1} - \frac{\partial \sigma_{ij}^{(1)}}{\partial x_1} \varepsilon_{ij}^{(2)} \right] q(x_\ell) w_\ell |J|_{x_\ell} \right) \tag{41}$$

with $INT, x_\ell,$ and $w_\ell$ being the total number, co-ordinates, and weights of sampling points for Gauss numerical integration for triangles. Meanwhile, $T_K$ in Figure 4 indicates the geometry transformation between the master element $\hat{\Omega}$ and the grayed triangle $\Omega_K$ and the master element $\hat{\Omega}$, which is defined by

$$T_K: x_\ell = \sum_{i=1}^{3} x_i \psi_i(\xi, \eta)_\ell, \quad y_\ell = \sum_{i=1}^{3} y_i \psi_i(\xi, \eta)_\ell \tag{42}$$

where $\psi_i$ stand for the constant-strain FE basis functions, which are employed for the test displacement field $v(x)$.

## 4. Numerical Experiments

The enriched NEM module was developed in Fortran and combined into the previously developed PG-NEM code [30] having the stress recovery module [36]. To demonstrate and validate the present method, the crack problem of an infinite plate with collinear cracks depicted in Figure 5a is considered. Differing from the cracks in plates of finite size which are of great practical interest, the present infinite plate with periodic cracks provides the closed form solution. Meanwhile, for the numerical study, one may take two kinds of periodic finite crack problems. One is obtained when the plate is cut along EF and GH, while the other is obtained if the plate is cut along AB and CD. Since the first one has been widely taken for the numerical studies (i.e., Ryicki and Kanninen [37] and Chow and Atluri [38]), the second periodic analysis model is taken for the current numerical experiments. Figure 5b represents the detailed second periodic model, as well as a NEM grid having higher grid density toward the crack tip for an upper left darkened quarter. The SIF for this infinite plate with collinear cracks is expressed in the following closed form (the correction factor C of 1.02) [39]:

$$K_I = \sigma_\infty \sqrt{\pi a} \left( \frac{2b}{\pi a} tan \frac{\pi a}{2b} \right) \cdot C \tag{43}$$

with *a* being the half crack length, where a circular path $\Gamma$ around the left crack tip has the relative radius $r/a = 0.0139$, and the near-tip stress distributions are to be evaluated along this circular path. The Young's modulus *E* is 200 GPa and Poisson's ratio *v* is 0.3.

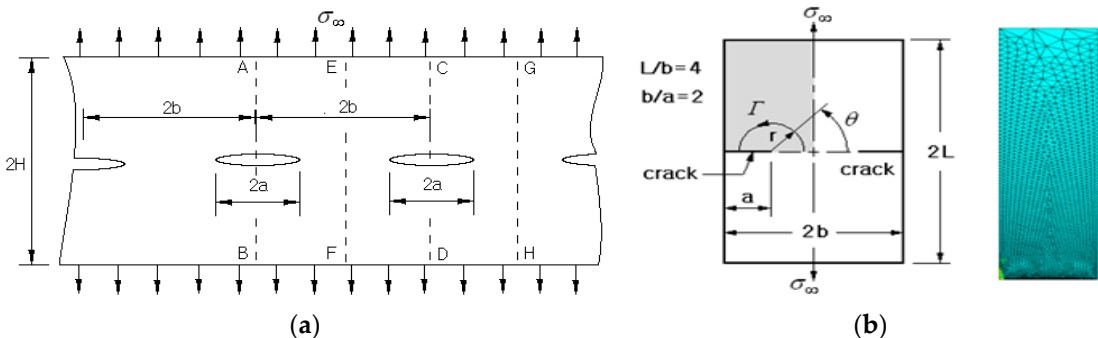

**Figure 5.** (**a**) Infinite plate with collinear cracks in plane strain state [2], (**b**) periodic model with double edge cracks and a gradient natural element method (NEM) grid (unit: m, $N = 2516$).

The problem was solved with 13 Gauss integration points, and the stress field approximated by PG-NEM was smoothened by the stress recovery technique [36] with the Laplace interpolation functions. The stress distributions were evaluated along the circular path $\Gamma$ and are compared in Figure 6. It can be observed that the case without enrichment shows similar stress distributions to the exact ones, but it provides a significantly low stress level. Thus, one can realize that only the fine gradient grid cannot sufficiently capture the near-tip stress distribution. Meanwhile, it is clear that the case with enrichment remarkably improves the stress level. Therefore, it has been justified that the enrichment of interpolation functions is effective in enhancing the near-tip stress approximation.

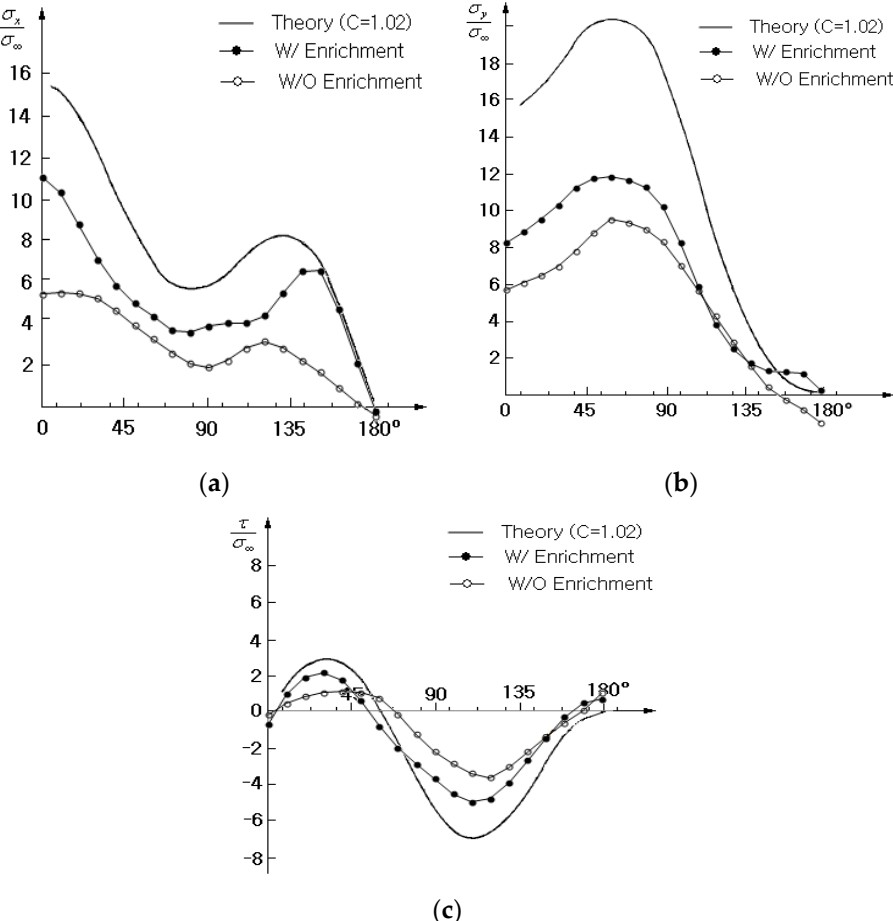

**Figure 6.** Comparison of the stress distributions along the path $\Gamma$ ($r/a = 0.0139$): (**a**) $\sigma_{xx}$, (**b**) $\sigma_{yy}$, and (**c**) $\tau_{xy}$.

Figure 7a shows a rectangular FGM plate with an edge crack which is in the plane strain condition, where $W$ and $H/W$ are 1.0 and 8.0 units, while $a/W$ is set variable for the parametric investigation as follows: 0.2, 0.3, 0.4, 0.5, and 0.6. The Young's modulus $E$ varies along the $x-$axis in a form of

$$E(x) = E_1 exp(\beta x), \quad 0 \leq x \leq W \tag{44}$$

but Poisson's ratio is uniform as 0.3. When $E_1$ and $E_2$ are specified to the elastic moduli at $x = 0$ and $x = L$, respectively, the parameter $\beta$ is calculated according to $\beta = ln(E_2/E_1)$, with $E_1$ being 1.0 unit. Two material variation cases of $E_2/E_1$ are taken as follows: $E_2/E_1 = 0.1$ and 10.0, and two loading cases of tension in Figure 7b and bending in Figure 7c are considered. This example was initially studied by Erdogan and Wu [40], who also provided an analytical solution. The external loading is set by as follows: for tension and $\sigma_{yy}(x, \pm 4) = \pm 1.0$ for tension and $\sigma_{yy}(x, \pm 4) = \pm(-2x + 1)$ for bending.

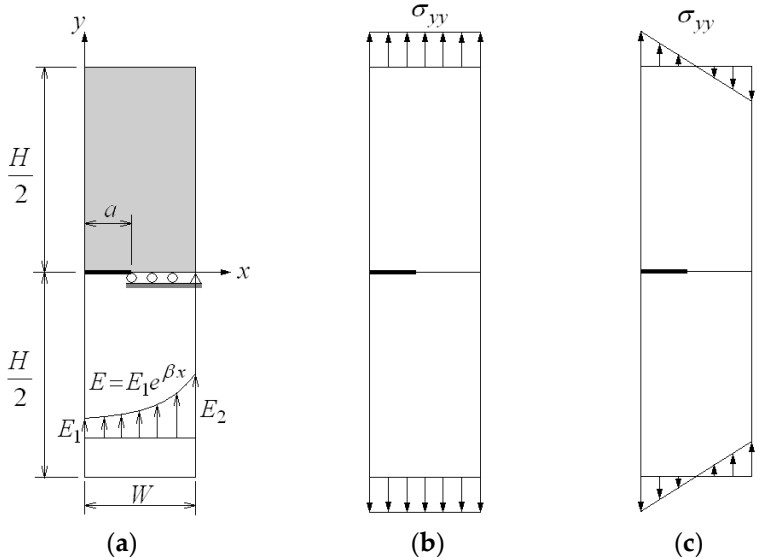

**Figure 7.** A cracked rectangular functionally graded material (FGM) plate with a varying Young's modulus: (**a**) Geometry dimensions, (**b**) tension loading, (**c**) bending.

An upper grayed half is taken for the crack analysis from the problem symmetry, and the following symmetric constraint is specified to the symmetric line. The non-cracked symmetric line is subjected to the vertical displacement constraint of $u_y = 0$, and the right end point is enforced by the lateral displacement constrain of $u_x = 0$. A $11 \times 41$ uniform NEM grid is used, and all the NEM analyses, the stress recovery, and the modified interaction integrals $\widetilde{M}^{(1,2)}$ were carried out with 13 Gauss points. The radius $r_{int}$ for determining the domain of integration $A$ was chosen using twice the side length of a square consisting of two Delaunay triangles [41].

Table 1 comparatively represents the normalized mode-I SIFs for uniform tensile loading and bending loading, respectively, where the values in parenthesis indicate the relative differences (%). It is confirmed that the proposed enriched PG-NEM provides the SIFs, which are consistent with the analytical solutions [40], for all the combinations of $E_2/E_1$ and $a/W$ of two types of loading, except for the relatively remarkable relative difference 6.807% in the tension case with $E_2/E_1 = 10$ and $a/W = 0.3$. For the further comparison, the numerical results by Kim and Paulino [15] and by Rao and Rahman [34] are also given in Table 1. It is observed that the SIFs of proposed method are correlated satisfactorily with the FEM results by the $J_k^*$-integral and the EFGM results by the modified $\widetilde{M}^{(1,2)}$-interaction integral. Table 2 represents the comparison with the results obtained without using the enrichment. In the uniform tensile loading, the case without enrichment provides the SIFs, which are lower than those with enrichment, such that the range of relative differences is from 27.59% and 51.04%. Meanwhile, in the bending, it is observed that the difference between with and without enrichment is not significant. Furthermore, the case without using enrichment provides the SIFs which are higher than those obtained by the enrichment. It is inferred that the near-tip stress field produced by bending is totally different from that by uniform tensile loading. The relative difference becomes smaller as the relative crack length $a/W$ increases. It is observed from the detailed numerical values that the relative difference ranges from 2.41% to 24.68%. Thus, it has been found that the proposed enrichment method is influenced by the loading type.

**Table 1.** Normalized stress intensity factors $K_I / \sigma_{yy} \sqrt{\pi a}$ for an edged cracked plate.

| | Method | $E_2/E_1$ | Relative Crack Length a/W | | | |
|---|---|---|---|---|---|---|
| | | | 0.2 | 0.3 | 0.4 | 0.5 |
| Uniform tension | Proposed Method | 0.1 | 1.302 | 1.843 | 2.557 | 3.504 |
| | | 10 | 0.960 | 1.150 | 1.581 | 2.176 |
| | Erdogan and Wu [40] | 0.1 | 1.297 (0.386) | 1.858 (−0.807) | 2.570 (−0.506) | 3.570 (−1.849) |
| | | 10 | 1.002 (−4.192) | 1.229 (−6.428) | 1.588 (−0.441) | 2.176 (0.000) |
| | Kim and Paulino $[J_k^*]$ [15] | 0.1 | 1.284 (1.402) | 1.846 (−0.163) | 2.544 (0.118) | 3.496 (0.229) |
| | | 10 | 1.003 (−4.287) | 1.228 (−6.352) | 1.588 (−0.441) | 2.175 (0.046) |
| | Rao & Rahman $[\widetilde{M}^{(1,2)}]$ [34] | 0.1 | 1.337 (−2.618) | 1.898 (−2.898) | 2.594 (−1.426) | 3.547 (−1.212) |
| | | 10 | 0.996 (−3.614) | 1.234 (−6.807) | 1.598 (−1.064) | 2.189 (−0.594) |
| Bending | Proposed Method | 0.1 | 1.885 | 1.872 | 1.908 | 2.141 |
| | | 10 | 0.583 | 0.636 | 0.812 | 1.069 |
| | Erdogan and Wu [40] | 0.1 | 1.904 (−0.472) | 1.886 (−0.742) | 1.978 (−3.539) | 2.215 (−3.341) |
| | | 10 | 0.565 (3.186) | 0.659 (−3.490) | 0.804 (0.995) | 1.035 (3.285) |
| | Kim and Paulino $[J_k^*]$ [15] | 0.1 | 1.888 (−0.159) | 1.864 (0.429) | 1.943 (−0.035) | 2.145 (−0.140) |
| | | 10 | 0.565 (3.186) | 0.659 (−3.490) | 0.804 (1.801) | 1.035 (3.285) |
| | Rao & Rahman $[\widetilde{M}^{(1,2)}]$ [34] | 0.1 | 1.903 (−0.420) | 1.875 (−0.160) | 1.954 (−2.354) | 2.155 (−0.650) |
| | | 10 | 0.564 (3.369) | 0.664 (−4.217) | 0.812 (0.000) | 1.045 (2.297) |

**Table 2.** Comparison of $K_I / \sigma_{yy} \sqrt{\pi a}$ between with and without enrichment.

| | Method | $E_2/E_1$ | Relative Crack Length a/W | | | |
|---|---|---|---|---|---|---|
| | | | 0.2 | 0.3 | 0.4 | 0.5 |
| Uniform tension | With enrichment | 0.1 | 1.3023 (46.902) | 1.8429 (29.041) | 2.5722 (37.785) | 3.5643 (27.587) |
| | | 10 | 0.9960 (45.050) | 1.2225 (47.706) | 1.5210 (47.843) | 2.1473 (51.041) |
| | Without enrichment | 0.1 | 0.6915 | 1.3077 | 1.6003 | 2.5810 |
| | | 10 | 0.5473 | 0.6393 | 0.7933 | 1.0513 |
| Bending | With enrichment | 0.1 | 1.8847 (−24.683) | 1.8719 (−12.132) | 1.9083 (−4.910) | 2.1406 (−2.406) |
| | | 10 | 0.5826 (18.641) | 0.6361 (7.436) | 0.8122 (9.554) | 1.0686 (15.319) |
| | Without enrichment | 0.1 | 2.3499 | 2.0990 | 2.0020 | 2.1921 |
| | | 10 | 0.4740 | 0.5888 | 0.7346 | 0.9049 |

The third example is a slant edge crack in a 2-D FGM plate in plane stress condition with the height $H = 2$ units and the width $W = 1$ unit. Referring to Figure 8a, the crack angle $\alpha$ is 45° and the relative crack length is $a/W$ is $0.4\sqrt{2}$, where the enrichment and the crack evaluation are performed within the inclined Cartesian co-ordinate system $\{x', y'\}$ originated at the crack tip $O'$. The Poison's

ratio is kept uniform as $\nu = 0.3$, but the elastic modulus varies in an exponential function along the $x$−direction

$$E(x) = \overline{E}exp[\eta(x - 0.5)], \quad 0 \le x \le W \tag{45}$$

Here, $\overline{E} = 1$ unit and $\eta$ is variable for the parametric study such as $\eta a = 0, 0.1, 0.25, 0.5, 0.75,$ and $1.0$. As external load, an exponentially varying distributed load is applied to the upper edge with $\sigma_{yy}(x,1) = \overline{\varepsilon}\overline{E}exp[\eta(x - 0.5)]$, with $\overline{\varepsilon}$ being 1. The whole bottom edge is constrained in the vertical direction (i.e., $u_y = 0$). At the same time, the right end is constrained in the horizontal direction. A uniform NEM grid, given in Figure 8b, is employed which is discretized by $11 \times 21$, where a darkened area indicates the domain A for the interaction integral. The total number of grid points is 235 because four additional grid points are needed along the crack to split a crack line into upper and lower faces. All the NEM analyses, patch recovery, and the modified interaction integrals were performed with 13 Gauss points.

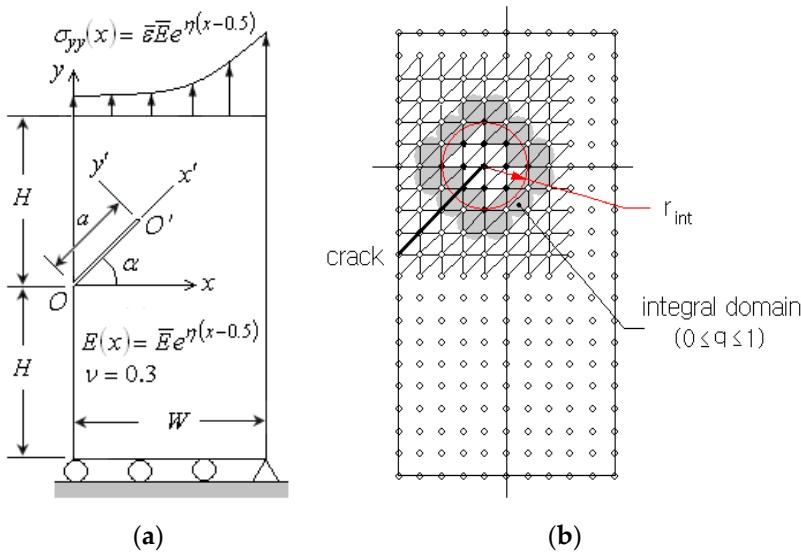

**(a)**                                **(b)**

**Figure 8.** (**a**) An FGM plate with an inclined edge crack under the exponential distribution load, (**b**) a uniform NEM grid and the domain of integration (235 nodes).

Table 3 comparatively represents the predicted normalized mixed-mode SIFs $K_I/\overline{\varepsilon}\overline{E}\sqrt{\pi a}$ and $K_{II}/\overline{\varepsilon}\overline{E}\sqrt{\pi a}$ evaluated by three different methods, the present PG-NEM, $J_k^*$−integral using FEM, and EFGM for seven different $\eta a$ values. It is confirmed that the present method is in good agreement with FEM and EFGM, such that the biggest relative difference is 4.37% at $K_{II}/\overline{\varepsilon}\overline{E}\sqrt{\pi a}$ for $\eta a = 0.25$. The values in [*] are the normalized mixed-mode SIFs obtained by PG-NEM without using enrichment. It is observed that the values are significantly smaller than those obtained using enrichment, such that the relative difference ranges from 23.32% to 65.63%. The relative difference increases in proportional to the exponential index $\eta a$. Hence, it was confirmed that the enrichment successfully and significantly improves the prediction accuracy of mixed-mode SIFs of the highly heterogeneous FGM, even for the coarse NEM grid.

**Table 3.** Normalized stress intensity factors (SIFs) for an inclined crack in a functionally graded plate.

| $\eta a$ | Present Method | | Kim and Paulino $[J_k^*]$ [15] | | Rao and Rahman $[\widetilde{M}^{(1,2)}]$ [34] | |
|---|---|---|---|---|---|---|
| | $\dfrac{K_I}{\bar{\varepsilon}E\sqrt{\pi a}}$ | $\dfrac{K_{II}}{\bar{\varepsilon}E\sqrt{\pi a}}$ | $\dfrac{K_I}{\bar{\varepsilon}E\sqrt{\pi a}}$ | $\dfrac{K_{II}}{\bar{\varepsilon}E\sqrt{\pi a}}$ | $\dfrac{K_I}{\bar{\varepsilon}E\sqrt{\pi a}}$ | $\dfrac{K_{II}}{\bar{\varepsilon}E\sqrt{\pi a}}$ |
| 0.0 | 1.449 [0.951] | 0.606 [0.354] | 1.451 (−0.138) | 0.604 (0.331) | 1.448 (0.069) | 0.610 (−0.656) |
| 0.1 | 1.377 [0.831] | 0.589 [0.335] | 1.396 (−1.361) | 0.579 (1.727) | 1.391 (−1.006) | 0.585 (0.684) |
| 0.25 | 1.277 [0.673] | 0.525 [0.266] | 1.316 (−2.964) | 0.544 (−3.493) | 1.312 (−2.667) | 0.549 (−4.372) |
| 0.5 | 1.163 [0.460] | 0.488 [0.241] | 1.196 (−2.759) | 0.491 (−0.611) | 1.190 (−2.269) | 0.495 (−1.414) |
| 0.75 | 1.087 [0.357] | 0.434 [0.159] | 1.089 (−0.184) | 0.443 (−2.032) | 1.082 (0.462) | 0.446 (−2.691) |
| 1.0 | 1.029 [0.240] | 0.411 [0.147] | 0.993 (3.625) | 0.402 (2.239) | 0.986 (4.361) | 0.404 (1.733) |

[*] indicate the SIFs obtained without using the enrichment, while (*) indicate the relative differences (%).

## 5. Conclusions

In this paper, an enriched PG-NEM was introduced for the reliable crack analysis of inhomogeneous functionally graded materials (FGMs). The Laplace interpolation function was enhanced by the crack-tip singular displacement field and the essential boundary condition was modified accordingly. The unreached global displacement field was extracted, and its stress field was smoothened by the stress recovery. The validity and effectiveness of proposed method was illustrated and justified through three representative examples.

Through the numerical experiments, it was found that enriched PG-NEM is more effective to represent the near-tip stress singularity. The enrichment provides the stress level, which is more close to the exact solution when compared to the use of locally refined NEM grid. In addition, it was observed that enriched PG-NEM accurately predicts the mixed-mode SIFs of inhomogeneous functionally graded materials with exponentially varying elastic modulus. When compared to the case without using enrichment, the prediction accuracy was improved up to four times. Meanwhile, with respect to the analytical solution, the maximum relative difference was found to be less than 6.5%.

**Funding:** This research was supported by Basic Science Research Program through the National Research Foundation of Korea (NRF) funded by the Ministry of Education (Grant No. NRF-2017R1D1A1B03028879).

**Conflicts of Interest:** The author declares no conflicts of interest.

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
