# Peer review of "A Numerical Evaluation of SIFs of 2-D Functionally Graded Materials by Enriched Natural Element Method"

_applsci, doi:10.3390/app9173581_

Round 1

Reviewer 1 Report

The Author focused on numerical prediction of stress intensity factors inhomogeneous functionally graded materials by an enriched Petrov-Galerkin natural element method. The overall trial displacement field has been approximated in terms of Laplace interpolation functions, and the crack tip one had enhanced by the crack-tip singular displacement field. According to Author the proposed method has been validated through the representative numerical examples, and the effectiveness has been justified by comparing the numerical results with theoretical ones. The article seems to be interesting for Readers. It creates a logical story and that is why in my opinion it could be published in "Applied Sciences". Some of the comments on the manuscript are listed below.

Line 92, the symbol t^ is called “the surface traction”, but in the title of this article it is marked that the problem is two dimensional. Please consider the possibility to change the name from already proposed by the Author “t^ the surface traction” to, for example, “the contour traction” or different which could suggests that the problem is two dimensional.

Line 105, in equation (6) the symbols: δ1j and nj’ are not defined (named). Please define these symbols.

There are differences between symbols presented in Figure 1 and in equation (6), for example (nt) and (nj’), respectively. The reviewer suggests unifying them.

Line 109, the smooth function q(x) is defined but for the Reader it is not clear why did this function appear in equation (7)?

Line 117, instead “integration” should be “integral”.

In equation (7) the symbol (nj’) disappeared and the sign has been changed. Could you explain why?

Equation (12), the new introduced quantities: W(1), W(2), W(1,2) should be named and the appropriate mathematical expressions for each of them should be assigned.

The passage from equation (11) to (13) is not clear. Some terms have been omitted. The additional explanation for Readers is required (especially why some partial expressions have been omitted?).

Line 258, there is small spelling mistake.

Please consider placing in the reference section some additional current literature position connected with finite element method, fracture mechanism as well as plastic and brittle cracking which overlap with Your field of interest as for example: Kaczmarczyk, J.; Grajcar, A. Numerical Simulation and Experimental Investigation of Cold-Rolled Steel Cutting. Materials 2018, 11, 1263.

In manufacturing plants many mechanical parts are made for example by gluing. Let’s consider two parts made of different kind of materials e.g. an aluminium and lead glued together and the cracking problem which occurs in the connection of those two different materials (along the border of two different materials). In this case Young’s moduli change discreetly. I wonder if your new proposed approach could be applied to solve such problem.

Author Response

I have attached a word file.

Reviewer 2 Report

The paper deals with an enriched Petrov-Galerkin natural element method (PG-NEM) to increase the numerical predictions of stress intensity factors, particularly for 2-D inhomogeneous functionally graded materials (FGMs). Numerical examples are provided for both homogeneous and inhomogeneous materials, to validate the proposed method. The proposed method is well developed and explained and numerical experimentation is convincing. However, the following observations must be addressed to the author:

The author makes extensive use of the terms "integral domain" and "integral region", while he presumably intends to mean "domain of integration". An integral domain is defined in abstract algebra as a nonzero commutative ring in which the product of any two nonzero elements is nonzero. Please review this term throughout the text.

Line 125: “The last term in equation (10) becomes extremely small as a contour G shrinks to the crack tip”. Is it actually the contour G or the contour C that shrinks to the crack tip? If contour G shrinks to the crack tip but contour G0 does not, the integration (not integral!) domain A can be reasonably large and this is in contradiction with what was said in the following sentence. The only way to have a narrow integration domain is to shrink both G and G0, that is, contour C to the crack tip.

There are 2 Figure 6. Please update captions and cross-references where appropriate.

Table 1 and Table 2: to improve the readability of these tables, add a column with the relative differences.

Line 319 and 322: “the case of without enrichment provides the SIFs which are lower than those of with enrichment” and “the case without using enrichment provides the SIFs which are higher than those obtained by the enrichment”. Lower or higher are not relevant. The author must focus attention on the relative differences, not on the lower or greater values.

Line 337: does the author mean “h” or “ha”? In line 338 and following text the author uses “ha”.

Line 340: “h” or “ha”?

Table 3: same observation as for Tables 1 and 2. Add the relative differences also for the values in brackets.

Line 357: “It is observed that the values are significantly smaller than those obtained using enrichment”. This is not relevant. The author must focus attention on the relative differences, not on the lower or greater values.

Author Response

I have attached a word file.

Reviewer 3 Report

Dear autor,
the article is very interesting and well written. I have only three comments. Some drawings are illegible: fig. 2, 3, 6a, 6b, 6, 7b. Formula number (16) should be at the end of the line. Line 301 here should by figure number 7 and line 350 fig. 8.

Author Response

I have attached a word file.
